# Interplay of Cardiometabolic Syndrome and Biliary Tract Cancer: A Comprehensive Analysis with Gender-Specific Insights

**DOI:** 10.3390/cancers16193432

**Published:** 2024-10-09

**Authors:** Vincenza Di Stasi, Antonella Contaldo, Lucia Ilaria Birtolo, Endrit Shahini

**Affiliations:** 1Center of Nutrition for the Research and the Care of Obesity and Metabolic Diseases, National Institute of Gastroenterology IRCCS “Saverio de Bellis”, Castellana Grotte, 70013 Bari, Italy; vincenza.distasi@irccsdebellis.it; 2Gastroenterology Unit, National Institute of Gastroenterology IRCCS “Saverio de Bellis”, Castellana Grotte, 70013 Bari, Italy; antonella.contaldo@irccsdebellis.it; 3Department of Clinical, Internal, Anesthesiology and Cardiovascular Sciences, Umberto I Hospital, Sapienza University of Rome, 00185 Rome, Italy; luciailaria.birtolo@uniroma1.it

**Keywords:** carcinogenesis, cholangiocarcinoma, gallbladder cancer, obesity, non-alcoholic fatty liver disease, metabolic dysfunction-associated steatotic liver disease, metabolic dysfunction-associated steatohepatitis, cardiovascular disease

## Abstract

**Simple Summary:**

Metabolic syndrome (MetS), metabolic dysfunction-associated steatotic liver disease (MASLD), and diabetes are all linked to Cholangiocarcinoma (CCA) in various ways. MASLD may have an increased risk of intrahepatic-CCA, whereas untreated patients with shorter diabetes durations were more likely to develop biliary tract cancer (BTC). More research is needed to understand how reproductive hormones cause BTC. BTC patients may be at increased intrinsic cardiovascular risk of neoplastic/non-neoplastic cardiac complications. Therefore, early detection/prevention of chronic liver disease, as well as intervention studies, will almost certainly be required to determine whether improvements in MetS, weight loss, and diabetes therapy can reduce CCA risk and progression.

**Abstract:**

BTC overall incidence is globally increasing. CCA, including its subtypes, is a form of BTC. MetS, obesity, MASLD, and diabetes are all linked to CCA in interconnected ways. The link between obesity and CCA is less well-defined in Eastern countries as compared to Western. Although more research is needed to determine the relationship between MASLD and extrahepatic CCA (eCCA), MASLD may be a concurrent risk factor for intrahepatic CCA, particularly in populations with established or unidentified underlying liver disease. Interestingly, the risk of biliary tract cancer (BTC) seemed to be higher in patients with shorter diabetes durations who were not treated with insulin. Therefore, early detection and prevention of chronic liver disease, as well as additional intervention studies, will undoubtedly be required to determine whether improvements to MetS, weight loss, and diabetes therapy can reduce the risk and progression of BTC. However, further studies are needed to understand how reproductive hormones are involved in causing BTC and to develop consistent treatment for patients. Finally, it is critical to carefully assess the cardiological risk in BTC patients due to their increased intrinsic cardiovascular risk, putting them at risk for thrombotic complications, cardiovascular death, cardiac metastasis, and nonbacterial thrombotic endocarditis. This review aimed to provide an updated summary of the relation between the abovementioned cardio-metabolic conditions and BTC.

## 1. Introduction

Cholangiocarcinoma (CCA), a kind of biliary tract cancer (BTC), is the second most common primary liver malignancy, after hepatocellular carcinoma (HCC), accounting for 1% of all neoplasms. Even though the geographic distribution of CCA varies substantially globally, the general incidence is expanding [1]. CCA develops at distinct sites along the biliary tree, and based on anatomical sites, CCA can encompass intrahepatic (iCCA) and extrahepatic (eCCA) forms [2], which determines the range of mutations that occur. Specifically, iCCA has been found to originate from hepatic progenitor/stem cells and mature hepatocytes/cholangiocytes, whereas eCCA can be traced back to both biliary progenitor/stem cells and mature cholangiocytes [1].

Notably, iCCA contributes to 10–15% of primary liver cancers, being divided into small and large duct types [2]. Furthermore, eCCA is classified as perihilar or distal, depending on whether it originates above or below the cystic duct [1]. Another predominant type of (BTC) is gallbladder cancer (GBC), which is primarily caused by prolonged chronic inflammation, regardless of whether it originates from lithiasis or non-lithiasis sources [3]. Figure 1 depicts the various anatomical positions of CCA and GBC.

The BTC subtypes under consideration exhibit differences not only in their anatomical location but also in their etiopathogenesis [1]. These distinctions have a paramount role in the biological underpinnings of CCA and are distinguished by unique risk factors and molecular landscapes, proposed cells of origin, and distinct genome aberrations. Interestingly, both a highly reactive desmoplastic stroma and a complex mechanism governing the mutual interactions of tumor cells and the stromal compartment are distinctive characteristics of iCCA [4]. A previous in vivo study revealed that the hepatic cell lineage, not the bile duct cell lineage, was the source of iCCA [5]. When it comes to the early stage of the disease for both types of CCA and GBC, surgical resection is typically the preferred therapy, aiming to be radical. Nevertheless, recurrence poses a significant risk in many cases, and adjuvant treatment has a low success rate.

## 2. Risk Factors

### 2.1. Biliary Tract Diseases and Concurrent Metabolic/Viral Infection Factors

A recent meta-analysis of CCA risk factors discovered that biliary cysts/stones, cirrhosis, hepatitis B, and hepatitis C were the most meaningful risk factors for iCCA and eCCA. In particular, choledochal cysts have the highest risk of iCCA and eCCA. In addition, the relationship between hepatitis B and iCCA differed between Eastern and Western populations. Certain risk factors, like excess body weight and diabetes, even if not strongly associated, are increasing worldwide and may contribute to rising CCA rates [6]. Notably, a robust correlation has been found between perihilar CCA and primary sclerosing cholangitis (PSC), implying that eCCA, including perihilar and distal cancers, tends to be associated with biliary tract diseases [7]. On the other hand, the finding that statin use was associated with significantly reduced risk for all eCCA (mediated by the lowering of the endogenous cholesterol synthesis) such as perihilar CCA and distal CCA subtypes could represent a surrogate for the contributing function of hyperlipidemia effects to the increased secretory process and toxicity within the bile ducts of excess cholesterol release, especially in a heterogeneous population of patients with liver–not-liver-specific comorbidities (e.g., diabetes, cholangitis, PSC, inflammatory bowel disease, and hepatitis C). Likewise, patients with distal CCA who employed statins had remarkably higher overall survival [8].

### 2.2. Metabolic Syndrome, Diabetes Mellitus, and Obesity

As previously mentioned, CCA and GBC are the prevalent subtypes of BTC. Many kinds of cancers, including BTC, are related to chronic inflammation and therefore appear to be a risk factor for these diseases. Recently, evidence suggests that one of the key mediators of cancer development and progression is the inflammatory process itself [9]. It is now established that metabolic syndrome (MetS) is associated with a chronic low-grade inflammation state involving several organs [10]. For this reason, it is easy to think of MetS as a risk factor for biliary oncogenesis. Accordingly, between potentially preventable causes of cancer, we can recognize excess body weight [11,12].

How excess body weight promotes carcinogenesis in the hepatobiliary tract is still not fully understood but it is certainly multifactorial. A simple classification divides potential mechanisms into either indirect (for example, excess body weight promotes gallstone formation, which then drives local inflammation-driven carcinogenesis), or direct, in which pro-tumorigenic cell signaling occurs as a consequence of excessive adiposity [13]. It is believed that Interleukin 6 (IL-6), tumor necrosis factor (TNF), and macrophage migration inhibitory factor (MIF), which are classic cytokines and growth factors synthesized and released by adipose tissue, have direct pro-tumorigenic properties in the gastrointestinal and hepatobiliary tract [13].

In a very large series, the authors analyzed data from the Biliary Tract Cancers Pooling Project (BiTCaPP), which consists of 27 studies (with over 2.7 million adults), including 22 prospective cohort studies, observational follow-up of participants enrolled in four randomized controlled prevention trials, and one cancer screening trial from different countries [14]. These authors found that for each 5 kg/m^2^ increase in body mass index (BMI), there were risk increases for GBC (HR = 1.27), iCCA (HR = 1.32), and eCCA (HR = 1.13), but not ampulla of Vater cancer (AVC) (HR = 0.99). Specifically, increasing waist circumference, hip circumference, waist-to-hip ratio, and waist-to-height ratio were associated with GBC and iCCA but not eCCA or AVC [14]. From this perspective, it can be concluded that weight management programs may help minimize the risk of these diseases.

In a Chinese case–control study, in contrast, the authors found that AVC was also related to diabetes, cholecystolithiasis, chronic pancreatitis, total cholesterol, high-density lipoprotein, and apolipoprotein A [15]. In another Chinese population-based case–control study, the authors analyzed the association between MetS/Insulin Resistance (IR) and BTC (specifically GBC and eCCA) [16]. In this study, MetS was strongly related to GBC (OR = 2.75), especially with an increase in the number of MetS components (*p* < 0.001), while the association between IR and GBC was borderline (*p* = 0.06) [16]. Notably, GBC is associated with the presence of gallstones [17]: in this study, MetS was also significantly associated with biliary stones (OR = 1.64) [16].

Regarding CCA, in a case–control series, obesity was identified as one of the significant risk factors associated with iCCA versus eCCA [18]. For both BTCs (GBC and CCA), another case–control study estimated an increased risk among obese people (OR = 1.52) [19].

A greater risk of BTC was also found in type 2 diabetes mellitus (T2DM), a condition characterized by IR such as MetS. In this regard, in a systematic review and meta-analysis by Ren et al. including 21 studies, DM was linked to an increased risk of BTC compared with no DM (summary RRs = 1.43), with significant heterogeneity among studies (*p* = 0.001). In this study, DM was also positively associated with the risk of GBC or eCCA, but not AVC [20]. In 2013, in a European prospective analysis, authors evaluated the associations between self-reported DM, its duration, age at DM diagnosis, and insulin treatment with the risk of BTC and HCC, independent of obesity [21]. After 8.5 years of monitoring, they observed 204 BTC cases (75 GBC and 129 non-GBC cases) and 176 HCC cases. In this study, a higher risk of BTC and HCC was associated with DM status, independent of anthropometric parameters such as BMI and waist-to-height ratio (RRs = 1.77 and 2.17, respectively). Participants with shortened diabetes periods and those who received no treatment with insulin had a greater probability of BTC. Concerning BTC subtypes, DM was only associated with GBC (RRs = 2.72) while the treatment with insulin was mostly related to the risk for HCC. No appreciable differences were found in HBV- and HCV-negative patients [21]. This study shows that diabetes, in addition to being associated with obesity, may contribute to BTC.

In a prospective Korean study, the authors evaluated the association of prediabetes, diabetes, and diabetes duration with the subsequent risk of BTC, including CCA and GBC [22]. Adjusted HR values for BTC of 1.08, 1.31, 1.35, and 1.47 were found for the impaired fasting glucose, newly diagnosed diabetes, diabetes duration less than 5 years, and diabetes duration higher than 5 years groups, respectively. Finally, worsening glycemic status was associated with a significant increase in the risk of BTC (*p* for trend < 0.001) [22].

Regarding CCA specifically, there may be some differences in risk and progression factors for both iCCA and eCCA. A 2017 Surveillance, Epidemiology, and End Results (SEER)-Medicare resource analysis evaluated risk factors and novel pre-existing medical conditions potentially associated with iCCA and eCCA [23]. In this study, similar risk factors were found for iCCA and eCCA, with the exceptions of thyrotoxicosis and hemochromatosis. Specifically, metabolic disorders such as metabolic dysfunction-associated steatotic liver disease (MASLD), obesity, and diabetes were associated with increased risks of both cancer types [23].

To assess the link between MetS and the development of HCC and iCCA, a previous analysis of the same database conducted in 2011 by Welzel et al. identified all subjects diagnosed with HCC and iCCA between 1993 and 2005. For comparison, a 5% sample of individuals living in the same regions as the SEER registries of the cases was selected [24].

The authors discovered that MetS was significantly more common among patients who developed HCC (37.1%) and iCCA (29.7%) than the comparison group (17.1%, *p* < 0.001) in this analysis, which comprised 3649 HCC cases, 743 iCCA cases, and 195,953 comparison individuals. MetS was still significantly associated with a higher chance of developing HCC or iCCA after controlling for major risk factors and demographic covariates (OR = 2.13, *p* < 0.001 for HCC, OR = 1.56, *p* < 0.001 for iCCA) [24].

In another Chinese case–control study, MetS was strongly related to a 1.86-fold elevated CCA risk [25]. Specifically, MetS showed a significative and positive correlation with CCA subtypes, with adjusted ORs of 0.35 and 0.29 for iCCA and eCCA, respectively (both *p* < 0.001); obesity was related to both iCCA and eCCA, with similar ORs, while diabetes was only related to iCCA (OR = 4.59), but not eCCA (OR = 0.97) [25].

Regarding the molecular process, the authors of an intriguing study sought to determine whether extracellular matrix alterations, both qualitative and quantitative, could trigger biliary carcinogenesis in MetS patients with iCCA. Osteopontin (OPN), tenascin C (TnC), and periostin (POSTN) were observed to be considerably more deposited in 22 iCCA with MetS undergoing surgical excision than in matching peritumoral regions. Furthermore, in comparison to 44 iCCA samples without MetS, OPN deposition was considerably higher in MetS-iCCAs. HuCCT-1 (the human iCCA cell line) showed a considerable increase in cell motility and the cancer-stem-cell-like phenotype upon exposure to OPN, TnC, and POSTN. The location and components of fibrosis in MetS-iCCAs were different from those in non-MetS-iCCAs both quantitatively and qualitatively. As a result, they suggest that OPN overexpression is a characteristic that sets MetS apart in iCCA [26].

In a recent narrative review, the authors reported some molecular mechanisms potentially related to diabetes/obesity and CCA [27]. Specifically, they remembered that, despite IR, T2DM patients may be affected by the anabolic effects of insulin and that insulin is found in bile and supports the cellular proliferation of cholestatic cells. Furthermore, in vitro studies have demonstrated that tumor cells in the gallbladder exhibit elevated levels of insulin receptors and insulin-like growth factor (IGF) and that these cells require consistent insulin stimulation to proliferate [28,29,30]. Patients with CCA have significantly higher levels of insulin and IGF-1, according to other studies [31,32]. Regarding obesity, they reported some effects of adiponectin and leptin in cancer progression (the levels of these two adipokines are inversely and directly related to obesity) and they analyzed the role of a pro-inflammatory cytokine such as IL-6 and TNF-α in cancer development [27]. Specifically, TNF-α stimulates NFκB, MAPK, and Akt signaling, which in turn triggers the release of matrix metalloproteinase-9 (MMP-9) and increases the invasiveness of CCA in vitro [33].

Considering MetS as a modifiable risk factor (with possible variations over time), in a Korean nationwide population-based cohort study, the authors examined a sizable cohort of 8,581,407 adults who underwent laboratory testing and anthropometric measurements in two consecutive national health screenings between 2009 and 2012, with the subjects being monitored until 2017 [34]. The participants were categorized into four groups: MetS-persistent, MetS-developed, MetS-improved, and MetS-free. The MetS-persistent group exhibited a consistently greater chance of CCA compared to the MetS-free group (*p* < 0.001). Even after adjusting for various factors (adjusted (a)HR = 1.07), the MetS-persistent status was substantially related to an elevated risk of CCA compared with the MetS-free status (HR = 2.8). In the fully adjusted model, neither enhanced nor newly discovered MetS were linked with CCA risk (aHR = 1.02, and aHR = 0.99, respectively) [34]. Therefore, we can hypothesize that strategies to improve MetS can contribute to improving the onset and progression of CCA. Given the high incidence of obesity, especially in the Western world, preventing and combating it is certainly necessary to minimize its consequences, including the increased oncological risk.

A 2013 narrative review provided a summary of knowledge linking obesity and CCA and underlined the importance of developing studies also on the impact of the extent and method of weight loss on the risk of CCA [35]. A more recent review evaluated the epidemiological and molecular associations between these two conditions [36]. Regarding the epidemiological data in particular, this study shows that there is a stronger correlation between obesity and CCA in Western nations but a weaker correlation in Eastern ones. Nonetheless, it should be remembered that obesity is less common in Asian countries than it is in American ones and that there is a different cut-off point to define obesity itself in the Asian population [37]. Notably, in endemic areas of Southeast Asia and the Western Pacific, liver fluke infection is one of the most well-known causes of CCA [38].

Of interest, adipose tissue aromatase is responsible for the transformation of androgenic precursors into estrogens [39]. Beyond hormone-responsive tumors, there is currently evidence that other tumor types can also respond to hormonal stimuli. There is some evidence to suggest that estrogens likely contribute significantly to CCA development, but more research is required to determine how estrogens connected to obesity affect the growth axes of CCA [36,40,41,42,43,44,45].

In obesity, an alteration of the normal production of adipokines, especially leptin and adiponectin, is also observed. Leptin receptors are extensively expressed in peripheral tissues that are metabolically active and are crucial for maintaining energy homeostasis. Additionally, normal cholangiocytes express leptin receptors, and malignant transformation increases their expression [46]. In a recent observational study, an Italian group found increased serum leptin levels in patients with CCA compared with patients with benign biliary diseases [47]. Moreover, leptin and its receptor also serve a crucial function in cholangiocarcinogenesis in animal models [47,48]. Unlike leptin, adiponectin levels are lower in obesity. This adipokine is known for its anti-tumor effect in several cancers thus it is conceivable that its reduced levels in obesity may be related to carcinogenesis [49,50].

A very recent study evaluated an adiponectin receptor agonist in CCA in vitro and in vivo. This agonist inhibited proliferation, migration, invasion, and colony formation and induced apoptosis in a time- and dose-dependent manner in vitro (*p* < 0.05), while in the treated group compared to the control group, it reduced the number of CCA and tumor volume, prolonged survival, and decreased metastasis and ascites (*p* < 0.05) [51]. These results greatly encourage further studies on CCA target therapies based on these molecules.

Regarding the link between DM and CCA, in a meta-analysis of twenty studies, the author found that, compared with individuals without diabetes, the pooled OR of CCA was 1.74 (*p* = 0.568 for heterogeneity) for patients with diabetes, iCCA (summary RR = 1.93, *p* = 0.037 for heterogeneity), and eCCA (summary RR = 1.66, *p* = 0.001 for heterogeneity) [52].

One of the most important effectors of the increased risk of cancer in patients with T2DM is insulin. T2DM is characterized by IR and compensatory hyperinsulinemia. However, the binding of insulin to its receptor has not only a metabolic but also a mitogenic effect. The mitogen-activated protein kinase (MAPK) pathway mediates the mitogenic effect of insulin and ultimately promotes cell proliferation. This pathway can be related to the mechanism of increasing cancer risk. Also, insulin can act via the IGF for its mitogenic effect. Increased insulin levels cause the degradation of IGF binding protein leading to the release of free IGF, which promptly acts on its receptor inducing cell proliferation. Moreover, high insulin levels in compensatory conditions with hyperinsulinemia can directly bind to the IGF receptor (IGFR), which normally has a low affinity for this hormone. The overall effects make DM patients with hyperinsulinemia prone to developing cancer [53].

In order to evaluate the associations of obesity and DM with iCCA, some authors conducted a pooled analysis of individual-level data from 13 US-based cohort studies.

Specifically, they conducted a meta-analysis examining the associations between obesity, diabetes, and iCCA risk [54]. In a meta-analysis of prospectively determined cohorts and nested case–control studies, obesity correlated with a 49% boosted iCCA risk (RR = 1.49; N = 4 studies), while diabetes was linked to a 53% raised iCCA risk (RR = 1.53; N = 6 studies) [54].

Focusing on GBC, the other type of BTC, it should be emphasized that it is a notoriously rare though lethal malignancy; since the symptoms are usually nonspecific, the diagnosis is often made at an advanced stage, with a poor prognosis. Over 80% of GBCs are adenocarcinomas that originate from the fundus (60%), body (30%), or neck (10%) of the gallbladder [55]. Chronic gallbladder inflammation, often related to cholelithiasis, is likely the first step of the carcinogenic process that leads to chronic irritation of the mucosa and dysplasia, terminating in malignant transformation. From what has been discussed about CCA, it is reasonable to suppose that the low chronic inflammation of MetS is also related to GBC.

In over 84,000 men and 97,000 women included in The Cancer Prevention Study II Nutrition Cohort, the mortality from GBC according to BMI had a RR of 1.8 (95% CI, 1.1 to 2.9) in obese men with a BMI of 30.0 to 34.9 kg/m^2^ compared to men with a normal BMI, while obese women (BMI, 30.0 to 34.9 kg/m^2^) had a RR of 2.1 compared to women with a normal BMI [56]. Finally, another meta-analysis confirmed that obesity is an associated risk factor for GBC, with an RR of 1.66 [57].

In terms of MetS components, a Chinese case–control study found that elevated serum triglycerides levels could be the most significant independent predictor of GBC risk together when combined with gallbladder stone disease, while IR might act as the first one in GBC not related to gallbladder stone disease. Furthermore, they initially advocated for the sharp rise of serum triglyceride levels as a potential diagnostic or prognostic biomarker of GBC with gallbladder stone disease [58].

In a meta-analysis of 31 prospective cohorts’ study, the authors, however, did not find a statistically significant association between triglyceride levels and GBC but indicated that MetS was significantly associated with many subtypes of gastrointestinal cancer risk. Specifically, MetS was associated with a raised GBC risk (RR = 1.37) [59].

In another large cohort study comprising 578,700 men and women, a composite MetS score, based on BMI, blood pressure, and circulating concentrations of glucose, total cholesterol, and triglycerides was significantly associated with GBC risk [60]. Further investigation of individual metabolic risk factors found that BMI and hyperglycemia were significantly linked to a higher chance of GBC [60].

Interestingly, in line with the aforementioned evidence about the association between triglycerides and GBC risk, another Chinese multicentric research study showed that dyslipidemia was associated with a 2.67-fold increase in the risk of malignancy in patients with gallbladder lesions and that it was an independent risk factor for malignancy, regardless of the presence of other risk factors and MetS components [61].

However, in the literature, some studies have evaluated the association of diabetes with GBC. In 2016, a meta-analysis of observational studies about the association between DM and GBC was published [62]. This work examined 20 studies and found that diabetics had a higher risk of GBC than non-diabetics (Summary RR = 1.56), regardless of gender [63]. This elevated risk was independent of smoking, BMI, or a history of gallstones, but alcohol use could be a confounding factor [62]. Finally, diabetes appears to reduce the survival rate of GBC patients. Another meta-analysis revealed a strong correlation between diabetes and GBC mortality (HR = 1.10; *p* < 0.001) [64].

Table 1, Table 2 and Table 3 show, in detail, the characteristics of studies examining the relationship between MetS, diabetes, and obesity with BTC.

### 2.3. MASLD/MASH

MASLD affects roughly 25% of the world’s population and may be deemed a modifiable risk factor for BTC. Even though the specific process underlying how MASLD leads to the development of CCA is still unknown, MASLD may either directly or indirectly favor cholangiocarcinogenesis by inducing liver inflammation, fibrosis, or cirrhosis [63,65].

The tumor ecosystem of CCA comprises cancer stem cells, tumor cells, stromal cells, the extracellular matrix, and a diverse array of signaling molecules. In MASLD, the liver generates and releases proinflammatory cytokines and prooxidative mediators, which can promote carcinogenesis via proliferation, anti-apoptosis, and angiogenesis [66,67]. These molecules involve elevated insulin, IGF, IL-6, TNF-α, and inducible nitric oxide synthase [66,67].

Over the last 50 years, annual iCCA incidences in Western countries have steadily increased for unknown reasons [7].

The simultaneous escalation of MASLD and iCCA may lead to a probable association between these two diseases. MASLD has exhibited a higher effect on CCA risk in the iCCA subtype than in the eCCA subtype, implying that HCC and iCCA share a common pathogenesis, as demonstrated by an earlier study by Banales JM et al. that uncovered iCCA to be more frequently associated with chronic liver diseases than eCCA [7].

Concerning diet impacts, in a recent mouse model study, researchers investigated how a high-fat diet can worsen cholangitis and enable the development of CCA (beyond HCC). It utilized a liver-specific E-cadherin gene (CDH1) knockout mice model for studying liver tumor progression. These animals develop unintentional inflammatory processes in the portal regions as well as periductal onion skin-like fibrosis, which mimics the pathological changes seen with PSC. The severity of cholangitis and the number of iCCA were remarkably more elevated in mice fed a high-fat diet than in normal-diet mice, likewise, displaying greater aggressiveness [68].

Moreover, among further causal factors, MASLD-associated gut dysbiosis can lead to carcinogenesis by causing leaky gut, microbe-associated molecular patterns, and bacterial metabolites [69,70]. For example, among 28 CCA patients, a distinct CCA-related dysbiosis was discovered independent of underlying liver-associated diseases. Specifically, Firmicutes levels were lower, whereas Bacteroidetes were higher in the biliary microbiota of CCAs than controls. The most typical genera in CCA’s biliary microbiota were Enterococcus, Streptococcus, Bacteroides, Klebsiella, and Pyramidobacter. In addition, Bacteroides, Geobacillus, Meiothermus, and Anoxybacillus genera levels strongly increased in CCA patients’ biliary microbiota compared to controls [71].

A 2007 study by Welzel TM et al. using data from the US Surveillance, Epidemiology, and End Results-Medicare database revealed that chronic MASLD was exclusively associated with iCCA and not eCCA (iCCA, *p* = 0.02; eCCA, *p* = 0.08). MASLD prevalence was higher in iCCA patients than cancer-free controls (0.3%, *p* = 0.03) [18].

According to a 2017 systematic review and meta-analysis, MASLD may raise the chance of developing CCA. The study comprised seven case–control studies from mixed Western and Asian countries, with 9102 CCA patients (5067 iCCA and 4035 eCCA) and 129.111 controls [65]. The prevalence of MASLD in CCA cases and controls varied from 0.8% to 44.1% and 0.3% to 18.8%, respectively. In this meta-analysis [65], MASLD and an increased risk of CCA were linked, with a pooled OR of 1.95. MASLD was associated with both iCCA (OR = 2.22) and eCCA (OR = 1.55). Specifically, the overall pooled adjusted ORs for all CCAs, iCCAs, and eCCAs were 1.97, 2.09, and 2.05, respectively. Nonetheless, the findings were relatively lower than previous research, which uncovered that the prevalence of MASLD in the general population was between 8% and 42% in Asian countries and between 11% and 45% in North America [72,73]. Yet, this meta-analysis did not accurately estimate the influence of steatosis, steatohepatitis, or metabolic dysfunction-associated steatohepatitis (MASH)-related cirrhosis on CCA risk because the various studies arbitrarily used dissimilar methods to diagnose MASLD [65]. Regardless, there is no way to rule out the possibility that cirrhosis contributed to the connection between MASLD and CCA.

Subsequently, a 2021 nationwide Korean study screened 8,120,674 patients to ascertain the relationship between MASLD and BTC risk (encompassing both CCA and GBC subtypes) [74]. MASLD was correlated to a greater risk of BTC compared to those without MASLD during the study’s median surveillance of 7.2 years (aHR = 1.28). The aHRs for the connection between GBC, CCA, and MASLD were 1.14 and 1.33, respectively [74]. The aHR for BTC increased with the fatty liver index (FLI) (*p* < 0.001), indicating a higher likelihood of MASLD presence. Similarly, the study uncovered that having both MASLD and diabetes increased the BTC risk by 47% (aHR = 1.47) [75]. Likewise, the study found a substantial relationship between MASLD and BTC risk in both obese and non-obese individuals. In particular, MASLD was linked to a raised risk of BTC in non-obese and non-abdominal obesity groups compared to obese and abdominal obesity [74].

Another 2021 meta-analysis by Corrao et al. found a strong link between MASLD and iCCA (OR = 2.19) but not with eCCA (OR = 1.48) [75].

Nowadays, to emphasize the importance of insulin resistance in the pathophysiology of metabolic dysfunction and liver diseases, it has been proposed that the terms Metabolic dysfunction-associated steatotic liver disease (MASLD) and metabolic dysfunction-associated steatohepatitis (MASH) should replace the old non-alcoholic fatty liver disease (NAFLD) and nonalcoholic steatohepatitis (NASH) nomenclatures, respectively, with slight alterations to their descriptions [76,77].

While PSC is also a typical risk factor for eCCA, research indicates a linkage between MASLD or MASH and iCCA [78]. Although the pathogenic mechanisms of MASLD-associated iCCA are unidentified, patients with MetS show OPN overexpression in the tumor stroma, a sialoprotein that preserves bone equilibrium and some immune system functions [18].

As cited above, in a study by Welzel TM et al. [24], the prevalence of MetS among patients with iCCA was 29.7%, significantly higher than the control group (17.1%, *p* < 0.001). In adjusted multiple logistic regression analyses, MetS remained significantly associated with a greater likelihood of the iCCA subtype (OR = 1.56, *p* < 0.001) [24].

To answer the question of what the relationship is between MASLD and cancer risk, Liu Z et al. evaluated recently 352,911 UK Biobank Caucasian subjects from a large-scale cohort study with MASLD (37.2%) and examined its associations with incident events (N = 23,345) in 24 cancers [79]. The multivariable Cox regression model revealed that, when compared to non-MASLD, MASLD was particularly associated with GBC (HR = 2.20), and liver cancers (HR = 1.81), including iCCA and HCC [79]. These relationships remained significant after controlling for waist circumference or BMI and the number of MetS components. The risk-increasing allele PNPLA3 rs738409 considerably increased the association between MASLD and the risk of liver cancer, including iCCA [79]. However, hepatic steatosis was discovered using the non-invasive FLI score rather than the gold-standard liver biopsy.

It is ambiguous how MASH influences the development of iCCA and the prognosis after surgery and whether the link between MASH and iCCA is causal or merely coincidental due to the increased MASLD incidence itself [80]. Similarly, it is critical to identify the constraints of preliminary studies that investigate the relationships between MASH and iCCA. The lack of histopathologic confirmation of MASH in the underlying liver, the inclusion of all patients with iCCA regardless of resectability or stage, the absence of tumor characteristics, and the lack of long-term outcome evaluation make it challenging to interpret the findings of these research studies [80].

In a 2013 French study, Nkontchou G et al. examined the histopathologic changes in the distant nontumoral livers of 57 consecutive patients with a peripheral iCCA over a 16-year period who did not have cirrhosis or bile duct disease [81]. The two most relevant histopathologic abnormalities discovered were macrovesicular steatosis (>10% of hepatocytes) in 66% of patients, including 19% with MASH, and hepatocyte iron overload in 38% of cases [81].

Similarly, Reddy SK et al. (2013) specifically examined MASH prevalence in patients with a surgically removable iCCA [80]. The authors found a prevalence of MASH of 20% among patients with resectable iCCAs. This study used an international multi-institutional database with data from 181 predominantly Caucasian patients who had undergone surgical resection for iCCA; 17.1% had definite or borderline MASH. Specifically, patients with MASH exhibited significantly higher rates of lobular inflammation, hepatocyte ballooning, and hepatic steatosis than the controls (*p* < 0.001) [80]. Of note, the MASH group had a higher rate of macrovascular (35.5% vs. 11.3%, *p* = 0.01) and vascular tumor invasions (48.4% vs. 26.7%, *p* = 0.02). Regardless, there was no difference in overall or recurrence-free survival after iCCA resection between patients with and without underlying MASH (31.5 vs. 36.3; 17 vs. 19.4 months, respectively) over a median follow-up of 19.1 months [80]. Nonetheless, some potential biases should be taken into account since the role of neoadjuvant chemotherapy in causing background steatohepatitis could not be ruled out, and there were cases of concurrent hepatitis B or hepatitis C in both groups, which may have affected the results [80].

In some cases, the etiology of the liver disease is unknown, as in the study by Welzel TM et al. (2016), who examined the Danish Cancer Registry for 14 years and found 764 patients with histologically confirmed iCCA; iCCA patients were more likely to have “unspecified cirrhosis” than age- and gender-matched controls, which could retain “burnt-out” steatohepatitis [82].

Kinoshita M et al. conducted a hospital-based case–control study in 2016 to display that histologically confirmed MASH could be a risk factor for iCCA evolution [83]. The prevalence of MASH was 17% (N = 15/88, iCCA patients), which is consistent with the findings from previous studies. The iCCA group had significantly higher levels of MASH (*p* < 0.01) and obesity prevalence (*p* < 0.04) compared to the control group among patients undergoing surgical resection for either iCCA (N = 34, without known risk factors) or a metastatic liver tumor (N = 69; control group) [30]. Multivariate analysis also revealed that MASH (with age and serum g-GT levels) were independent risk factors for iCCA. The iCCA group with MASH had a significantly higher prevalence of hepatic fibrosis than the control group (*p* < 0.01), suggesting that hepatic fibrosis generated by MASH may play a role in the development of intrahepatic cholangiocarcinogenesis [83]. The carcinogenic process involved in iCCA onset in MASH patients may be analogous to that of HCC, though the mechanism is still unexplored.

Table 4 shows, in detail, the characteristics of studies examining the relationship between MASLD and BTC.

Figure 2 illustrates pathological factors linking MetS, MASLD, T2DM, and other chronic liver diseases to BTC and GBC.

## 3. Gender Factors and Biliary Tract Cancers

GBC and BTC incidence rates vary between males and women [84]. GBC is predominately female, with a global female-to-male incidence rate ratio of 2:1 [85]. In contrast, men globally had greater incidences of iCCA, eCCA, and AVC [84,85]. This gender distribution shows that reproductive hormones may be critical factors in gallbladder cancer development. Indeed, the gallbladder tissue expresses estrogen and progesterone receptors [86].

High parity is associated with gallstones, which are commonly a precursor to gallbladder dysplasia. During pregnancy, gallbladder volume rises, bile flow is reduced, and increasing estrogen levels cause raised cholesterol saturation of the bile. Progesterone promotes biliary stasis by weakening the smooth muscle contractility of the biliary system, resulting in the development of cholesterol gallstones [85,86,87]. Thus, repeated exposure to heightened levels of female reproductive hormones throughout many pregnancies may result in gallstone development, serving as a potential cause of increased GBC risk [85].

Gallstones are also a risk factor for CCA, and studies indicate that estrogen stimulates cancer formation in the biliary tract [85,86,87,88]. However, because these malignancies lack a strong female predominance, the significance of female reproductive factors in carcinogenesis throughout the biliary system remains unknown. In addition, given that the gender ratio of GBC in East Asia is approximately one, the impact of female reproductive hormones can differ by geographic region [88].

Data from 19 studies on over 1.5 million women getting involved in the Biliary Tract Cancers Pooling Project was examined distinctly for Asian and non-Asian women. High parity was associated with a greater likelihood of GBC (HR of ≥ 5 vs. 0 births: 1.72) and age at menarche (HR per year increase = 1.15, in Asian women, while reproductive years were correlated with GBC risk.

A recent pooled analysis of the Asia Cohort Consortium found that later menarche age was linked to iCCA (HR = 1.19) and eCCA (HR = 1.11) in Asian women exclusively [89]. Furthermore, Asian women reach menopause at a younger age than European or North American women, and they have seen a secular transition in reproductive patterns [90,91].

The effect of reproductive hormones was evaluated by examining the expression patterns of estrogen and progesterone receptors (ER/PR) in GBC tissues [92,93,94,95]. Because of the small number of studies and the wide range of quantitative methodologies employed to evaluate receptor expression, the results were inconsistent [92,93,94,95].

A group of Indian scientists conducted a prospective study of estrogen/progesterone receptor (ER/PR) expression in GBC and attempted to correlate receptor expression with patients’ clinicopathological profiles to understand its implications. ER and PR are expressed in a considerable proportion (23.4%) of GBC patients, most of whom had both (73%). Receptor expression correlates with metaplasia, dysplasia, and the early/operable stage of the tumor, but its absence connects to the inoperable/metastatic stage [95]. Hryciuk B et al. investigated the level of expression of female reproductive hormone receptors (ERα, ERβ, and PR), connective tissue growth factor (CTGF), and HER2 in GBC and the nearest normal tissue to determine their prognostic value. Both GBC and normal tissue showed no expression of ERα. Five GBCs (10.4%) had strong (3+) HER2 expression. There was a positive connection between HER2, CTGF, and ERβ expression in GBC and matched normal tissue.

A multivariate analysis found that patient age over 70 years, tumor size, and ERβ expression in GBC were significantly associated with poor prognosis (*p* = 0.003) [96]. According to these data, the MyPathway study, which included seven HER2-positive BTC patients treated with combining anti-HER2 antibodies trastuzumab and pertuzumab had an objective response rate of 29% [97]. In the NCT02675829 clinical study, 17% of HER2 amplified BTC patients responded to ado-trastuzumab emtansine [98]. The basket study demonstrated the efficacy of neratinib, a pan-HER tyrosine kinase inhibitor, in HER2-mutant BTC patients [99].

Additionally, circulating estrogen hormone levels in patients with iCCA were studied. Interestingly, a 2020 histology study of fifty-six post-menopausal women with iCCA from five prospective international cohorts found that doubling estradiol was associated with a 40% increased risk of iCCA (OR = 1.40) but not HCC (OR = 1.12). A doubling of 4-androstenedione concentration was associated with a 50% reduction in liver cancer risk (OR = 0.50), while sex hormone-binding globulin (SHBG) was associated with a 31% increase (OR = 1.31) [45]. Patients with iCCA test positive for ER-α and ER-β subtypes [100,101]. 17β-estradiol promotes neoplastic cell proliferation by upregulating ER-α and downregulating ER-β [100].

Estrogens can also regulate COX-2 synthesis by playing a crucial role in cholangiocarcinoma cell development [101]. Kilander and colleagues found a slightly higher risk of these cancers when women’s parity increased [102]. Moreover, a Thai in vitro study analyzed the effect of estrogen on cholangiocarcinoma cell proliferation and invasion in dose-dependent manners, being potentially inhibited by tamoxifen [103]; this phenomenon could suggest a further treatment possibility for CCA by hormonal regulation; however, as with GBC, further large-scale population studies are required [104].

Despite the limitations of cell line experiments, an Italian study found that VEGF may play a significant role in the proliferative effects of estrogens on human cholangiocarcinoma, and strategies based on ER and/or VEGF antagonism could aid in the delay of cancer progression [43]. Xiao-Mei et al. found that diosgenin, a steroid hormone precursor, had an essential role in inducing apoptosis in cholangiocarcinoma cells by increasing the levels of cytosol cytochrome C, cleaved-caspase-3, cleaved-PARP1, and the Bax/Bcl-2 ratio. Thus, diosgenin could be considered a potential target for pharmacological therapy [103].

Finally, data from North American cohort studies (the Liver Cancer Pooling Project, LCPP) and the UK Biobank from 1980 to 1998 and 2006 to 2010 revealed that hysterectomy was linked to a doubling of iCCA risk (HR = 1.98,) when compared to women aged 50–54 (natural menopause). Long-term oral contraceptive use (>9 years) was linked to a 62% rise in iCCA risk (HR = 1.62). Furthermore, there was no association between iCCA risk and other exogenous hormones or reproductive factors [44].

Ultimately, an interesting Korean study found some gender differences in the effect of MetS on the development of some forms of cancer [105]. In women, a high-risk metabolic profile was associated with a significantly higher risk of BTC (HR = 2.05) [105].

## 4. Cardiovascular Risk

Cardiovascular disease and cancer are the main causes of death worldwide. Although they are commonly considered separate entities, they have several points in common. First of all, similar risk factors (e.g., smoking, obesity, diabetes mellitus, arterial hypertension, alcohol abuse, inappropriate diet, sedentary life, sexually transmitted infections, and, not least, pollution) contribute to the development of both diseases. Inflammation links risk factors, cardiovascular diseases (CVD), and cancer [106].

Atherosclerosis is defined as the lipid accumulation in the vessel’s intima, where inflammation plays a role in all its stages, from the parietal adhesion of leukocytes to the production of cytokines by macrophages/monocytes to thrombotic complications. On the other hand, a chronic inflammatory state is responsible for about 10–20% of tumors [107,108].

The changes in lifestyle, including a Mediterranean and vegetarian diet, associated with reduced alcohol consumption, smoking cessation, and physical activity are the leading measures to be put in to reduce the incidence of cancer and CVD and their relapses [109,110,111,112].

Hyperglycemia, hyperinsulinemia, and increased factor of IGF-1 stimulate, on the one hand, the migration and proliferation of smooth muscle cells and the mechanism at the base of atherosclerosis, and on the other hand, insulin receptors hyper-expressed on cancer cells, stimulating their proliferation and diffusion [113].

Regarding the connection between arterial hypertension and cancer, high levels of angiotensin II cause vasoconstriction, stimulating endothelial growth factor and inducing neoangiogenesis, contributing to tumor development. Other hypothesized mechanisms are oxidative stress at the vascular level and a proinflammatory state [114].

Moreover, many drugs used in cardiovascular prevention (such as aspirin, statins, metformin, and beta blockers) have revealed promising pleiotropic properties in vitro and in vivo and have also been shown, in observational studies, to be effective at preventing certain neoplasms, specifically BTC [115,116,117]. In contrast, a survival analysis of 1140 patients with BTC suggested no survival benefit in patients with advanced BTC from concomitant use of ACE-I/angiotensin receptor blockers, aspirin, and statin with chemotherapy, generally or when analyzed by disease subtype and in combination with specified systemic therapies. However, they did not evaluate dose– and duration–response relationships, thus potentially diluting the survival benefit of higher doses or longer-term use [118].

Patients with cancers also have a hypercoagulation status due to factors including tissue factor, cysteine protease, tumor hypoxia, tumor-induced inflammatory cytokines, and carcinoma mucin.

Based on the abovementioned hypercoagulation status, cancer patients, including patients with BTC, are at risk of venous thromboembolism (VTE), including deep vein thrombosis and pulmonary embolism [119] and new-onset coronary artery disease [120].

Notably, in patients with gastrointestinal cancers, procoagulant proteins secreted by tumor cells may increase the risk of thrombus formation by altering the host system, creating a hypercoagulation status, and activating the coagulation cascade or platelets [119]. Regarding the clinical significance of VTE, major vessel invasion is related to the occurrence of VTE in advanced CCA. VTE considerably reduces overall survival and is an important unfavorable prognostic factor for survival [120].

In a 2012 study conducted by Jeon et al., advanced-stage C-reactive protein and treatment with chemotherapeutic agents were associated with the occurrence of VTE in patients with CCA. So, this study also confirms that VTE was an independent negative prognostic factor for survivors of CCA [121].

Yuan et al. conducted 2024 prospective cohort research with 485,936 UK Biobank participants free of baseline VTE to investigate the correlations of 21 gastrointestinal diseases with the likelihood of incident VTE. This study reveals that patients with GBC and BTC had a 50% increased chance of developing VTE [118]. Pfrepper et al. studied 133 patients with CCA and discovered that the ONKOTEV score and serum CA 19-9 levels are independently associated indicators of thromboembolic events [122].

Trousseau syndrome is known as thromboembolism caused by malignancy. Yamane et al. reported an uncommon case of multiple ischemic strokes caused by nonbacterial thrombotic endocarditis (NBTE) in a patient with GBC [123].

Endothelial damage with subsequent exposure of subendothelial tissue to circulating platelets, immune complex deposition, hypoxia with resulting plasma tissue factor abnormalities, hypercoagulability possibly due to hypo- and hyperandrogenemia, thrombocytopenia, decreased factors V, VIII, and XII, and antithrombin III, and increased fibrinolysis are all possible causes of NBTE. In up to half of the cases, NBTE begins with an embolic stroke.

NBTE primarily affects left-sided heart valves in people with BTC. Transesophageal echocardiography is the recommended imaging method to detect the vegetation associated with NBTE, albeit the final diagnosis can only be made through histological examination [124].

Also, it is reported that cancer patients, including patients with BTC, have an increased risk of CVD. Chen et al. (2022) conducted a competing risk analysis of cardiovascular death in 5925 patients with GBC, 247 of whom died from CVD. The analysis revealed that age, marital status, cancer cell differentiation, chemotherapy status, and year of diagnosis were risk factors for cardiovascular death in patients with primary GBC [125].

Xia et al. reported that the proportion of non-cancer fatalities gradually increases over time following a GBC diagnosis, with CVD and infectious illnesses being the most common causes. GBC patients are more likely to die from CVD, infectious diseases, other causes of death, and gastrointestinal problems [126].

Finally, rare cases of cardiac metastasis are reported in patients with GBC [127,128]. Metastases from GBC commonly affect the liver and the paraaortic lymph nodes and rarely the heart. Cardiac metastasis of GBC may occur via both hematogenous and lymphatic routes, with lymphatic routes being the most frequent.

Inoue et al. (2005) reported several metastases upon autopsy, including cardiac metastases, in the case of a 68-year-old woman diagnosed with advanced GBC [128]. Another old report (1960) stated that 5 patients in 122 cases of cardiac metastasis had a primary GBC [129]. Ultimately, only four recorded cases of cardiac metastasis of GBC were found in the Japanese literature [127,130].

## 5. Metabolomics and Biliary Tract Cancers Screening/Prognosis

Metabolomics is a new and attractive technique that combines high-throughput analytical methodologies to analyze low-molecular-weight metabolites in various biological fluids. It can be utilized for screening for biomarkers and diagnosing diseases [131]. Technologies using nuclear magnetic resonance spectroscopy or liquid chromatography/mass spectrometry-based metabolomic analysis are promising methods for distinguishing between BTC and benign biliary tract diseases.

Bile acids and salts are gaining more attention than other endogenous metabolites in the human body due to their tumor-promoting properties [132]. Examining the molecular composition of bile can provide essential mechanistic data on the pathological changes in biliary epithelia, as well as the determination of novel biomarkers released into the bile or serum by tumor cells that may serve as BTC early detection and/or prognostic determination [133].

Various metabolite profiling investigations of human bile have been carried out lately. Early studies demonstrated that the levels of phosphatidylcholine, bile acids, and other biliary lipids could accurately distinguish CCA patients from those with benign biliary diseases [134]. Patients with BTC had significantly lower levels of lysophosphatidylcholine, phenylalanine, 2-octenoylcarnitine, and tryptophan, and higher levels of taurine- and glycine-conjugated bile acids compared to patients with benign biliary tract disease [133]. In another recent study comparing metabolomes in duodenal juice from malignant and benign diseases, acetone was a significant predictor, with higher concentrations in the malignant group. Acetone had a comparable predictive value to serum CA19-9 levels (AUC, 0.73 vs. 0.69, *p* = 0.70) [135]. Moreover, a Chinese study showed that a bile acid panel with chenodeoxycholic acid and taurochenodeoxycholic acid was able to distinguish between CCA and benign biliary diseases, with a sensitivity of 85.7%, specificity of 93.3%, and AUC of 0.97. As a method of diagnosis for CCA, the specificity and sensitivity of this model were higher compared to that of presently utilized biomarkers for CCA [136]. Also, the combined use of the tauroursodeoxycholic acid and glycoursodeoxycholic acid plasma–stool ratios may differentiate patients with iCC from those with HCC, liver cirrhosis, and healthy adults, with AUC values of 0.80, 0.87, and 0.91, respectively [137].

If a biomarker is detected and validated in bile, it can be tested in more accessible fluids such as serum, urine, or plasma. These fluids can also be used for direct metabolomics studies. In certain studies, serum metabolomics also revealed promising diagnostic biomarkers [138,139]. A mass spectrometry-based profiling platform in the serum of patients identified an interesting metabolomic signature based on four distinct metabolites (21-deoxycortisol, bilirubin, lysoPC(14:0), and lysoPC(15:0)) that could be used to diagnose CCA with high accuracy, and this signature could distinguish between iCCA and eCCA [138]. Conversely, one UK study found no differences between serum profiles of individuals who had benign biliary strictures and CCA [140]. Raj et al. (2017) identified eight serum metabolites as promising biomarkers for the early detection of GBC and CCA. The study also found significant fluctuations in lactate and formate levels, indicating their potential as markers for chronic cholecystitis progression to GBC [141]. Additionally, an international study with biopsy-proven iCCA, HCC, or PSC and healthy individuals found that several metabolites had higher diagnostic capacity for iCCA than CA19-9. An algorithm combining serum glycine, aspartic acid, and sphingomyelins resulted in accurate discrimination between iCCA and HCC with a sensitivity of 75% and specificity of 90%. Notably, these results were further validated in an independent cohort [139].

Patients with CCA were studied using urine samples to identify potential biomarkers [131]. The study discovered some metabolic differences in the urine of CCA patients compared to the control group. A set of urine metabolites, including 3-methylhistidine, citric acid, cytosine, and others, were identified as potential biomarkers for primary eCCA [142].

On the other hand, only a few studies have demonstrated the ability of metabolites to predict recurrence or overall survival [131]. In a study involving 108 iCCA patients, a LASSO-Cox prediction model was developed. This model consists of ten survival-related metabolic biomarkers and has shown promise as a valuable tool for predicting the overall survival of iCCA patients after surgical resection. It can be used to help guide the selection of the most effective therapy for these patients [143].

Despite the encouraging results for some intriguing metabolites, metabolomic studies in CCA are still in their early stages. Yet, the transition to clinical use is not expected to occur quickly due to the difficulty of identifying specific diagnostic or prognostic metabolites amidst numerous patient-related confounding factors. Finally, the reproducibility and validation of these studies pose a genuine concern, primarily attributable to the utilization of diverse analytical platforms and sample preparation protocols. This necessitates the establishment of standardized processes to ensure consistent and reliable outcomes across future studies.

### Single-Cell RNA Sequencing and Biliary Tract Cancers

Communication among individual cells plays a crucial role in the exchange of information, and the polarity of a single cell is intricately associated with its functionality and capacity for proliferation [144]. CCA exhibits heterogeneity across genetic, epigenetic, and functional levels, contributing to its aggressive nature and presenting challenges in treatment [145].

Single-cell RNA sequencing (scRNA-seq) technology is gaining popularity in cancer research today. This advanced methodology enables the comprehensive analysis of tumor tissues at the individual cell level, thereby yielding crucial insights into the genomics and epigenetics of tumor cells [145]. This technology encompasses multiple stages, including tissue or organ separation, single-cell extraction, lysis of cells, RNA extraction, cDNA synthesis, library construction, single-cell sequencing, and subsequent data analysis [146]. The use of this technology for dissecting CCA has provided a unique perspective into CCA phenotype and functional diversity, such as inter/intra-tumor heterogeneity, malignant–stromal interactions, novel subtypes, treatment-induced changes, biomarkers for diagnosis or prognosis, and possible therapeutic targets [145].

Notably, a dynamic change in cancer stem-like cell/stromal/immune cell composition was discovered across CCA formation time points in a mouse model [146]. Of note, tumor-infiltrating immune cells are connected to CCA prognosis [146].

Certainly, elevated levels of transcriptomic differences have been linked to aggressive tumors and poor survival rates. Previous scRNA-seq analyses of iCCA tumor cells revealed significant intertumoral heterogeneity. The tumor cells of iCCA shared activated signaling pathways, such as IL-6/STAT3, Wnt, TGF, and TNF [147]. Zhang M et al. (2020) revealed that the malignant subclusters exhibited a high degree of patient specificity, indicating substantial molecular and transcriptomic intertumoral heterogeneity among iCCA samples [147]. Four distinct subsets of tumor cells were identified based on gene signatures such as the epithelial–mesenchymal transition, cell cycle and hypoxia, the interferon response, and high levels of serine peptidase inhibitor Kazal type 1 (SPINK1). This study pinpoints novel pathways that could be essential in CCA progression, particularly the SPINK1, a kinase inhibitor of premature trypsin activation that was connected to cancer stemness and an unfavorable outcome [147]. Moreover, there were distinct immune and stromal cell populations identified, particularly tumor-infiltrating lymphocytes and cancer-associated fibroblasts (CAFs), within the context of extensive inter- and intra-tumor heterogeneity. Although based on a limited sample size, this study advances the field by presenting an scRNA-seq transcriptomic scenery of iCCA.

In a separate study, Song et al. (2022) conducted scRNA-seq on iCCA tissues to shed light on the diversity of tumor cells [148]. The authors used scRNA-seq on CCA cells from fourteen patients with iCCA and non-tumor liver tissues to identify S100P and SPP1 as markers for various types of iCCA. Furthermore, the S100P-SPP1+ iCCA peripheral small duct type showed more SPP1+ macrophage infiltration, less aggressiveness, and higher survival rates than the S100P+ SPP1- iCCA perihilar large duct type. They also found that the transcription factor CREB3L1 controls S100P expression and promotes tumor cell invasion [148]. The authors also investigated immune cells in iCCA and found significant differences in infiltrating immune cells between patients and tumors versus adjacent normal tissues. Tumor tissues contained CD4+ T lymphocytes, macrophages, and Tregs, whereas non-tumor samples contained primarily MAIT cells.

Cancer stem cells are also recognized for their role in long-term tumor maintenance, as well as tumor initiation, recurrence, metastasis, and treatment resistance. Bian et al. utilized the established CytoTRACE computerized method to analyze scRNA-seq data to investigate human iCCA. The study confirmed cell malignancy through copy number variations analysis and revealed significant heterogeneity in stemness or differentiation states among the malignant cells in iCCA [149]. Particularly, malignant cells exhibiting elevated stemness had substantially fewer levels of major histocompatibility complex II molecules than low-stemness malignant cells, which expressed significant levels of cytokines such as CCL2, CCL20, CXCL1, CXCL2, CXCL6, CXCL8, TNFRSF12A, and IL6ST, implying that high-stemness malignant cells have an intrinsic mechanism for immune escape [149]. Alvisi G et al. (2022) found a high infiltration of hyperactivated CD4+ Tregs in iCCA tumors, as well as reduced CD8+ T-cell effector functions [150].

On the other hand, Li et al. used scRNA-seq profiling to study tumor heterogeneity in eCCA, comparing treatment-naïve eCCA tissues to normal bile duct tissues from eCCA patients. The study identified eleven distinct subclusters, which included various cell types and subtypes [151]. The malignant epithelial cells (M1-M5) displayed a wide range of characteristics and gene expressions, demonstrating the significant heterogeneity of eCCA [151]. However, in another study, eCCA had less interpatient heterogeneity than iCCA, with seventeen genes upregulated in eCCA and forty-five genes significantly upregulated in iCCA. According to gene set enrichment analysis, iCCA is involved in the DNA repair, G2/M checkpoint, complement, and fatty acid metabolism pathways. Moreover, eCCA demonstrated enrichment in the Notch and UV signaling pathways [145]. The authors found that exhausted cytotoxic CD8+ T lymphocytes expressed both cytotoxic and exhaustion markers, including LAG3 and TIGIT. The study found inactivated NK cells in adjacent and cancer tissues, suggesting that cytotoxic CD8 + T lymphocytes were the primary effectors.

Growing proof from genetic profiling and protein data analysis indicates distinct roles for CAFs in CCA growth and spread. CAFs can inhibit immune cell activity and form barriers that make it difficult for drugs and immune cells to enter the tumor [145]. Two human iCCA samples were analyzed using scRNA-seq, and the most abundant type was vascular CAFs, which were found primarily in the tumor core and microvascular region. These CAFs were distinguished by microvasculature, proliferation signature genes, and highly expressed inflammatory chemokines such as IL6 and CCL8. Also, vascular CAFs have been shown to actively interact with tumor cells via the pro-invasive IL6/IL6R axis, promoting cancer stemness and contributing to iCCA progression [147]. Recently, a study revealed the critical role of fibroblast–oncocyte interaction in the remodeling of the immunosuppressive microenvironment in iCCA patients. As a result, it may activate downstream signaling pathways such as PI3K-AKT and Notch in tumors, thereby initiating tumorigenesis [152]. On the other hand, a prior investigation identified two distinct CAF subpopulations in iCCA and discovered that inflammatory CAFs promote iCCA via hepatocyte growth factor interaction with tumor cells, inducing MET expression, whereas myofibroblastic CAFs promote iCCA via Has2/hyaluronic acid [153].

These findings have demonstrated the differences between iCCA and eCCA at the single-cell level, but more research is required for validation and exploration to provide more compelling evidence. Additionally, the integration of scRNA-seq with other high-throughput and high-resolution single-cell technologies will constitute a major breakthrough in comprehending the modified dynamic spatial interactions among tumor cells and immune system cells in CCA.

## 6. Conclusions

CCA is a malignant neoplasm with a dismal prognosis, and its incidence has risen over the past decades. Despite technological advances in revealing the pronounced heterogeneity of CCA, the differences between iCCA and eCCA remain incompletely understood.

MetS, MASLD, obesity, and diabetes are, in different but intersected ways, related to CCA. Although more research is necessary to determine the relationship between MASLD and eCCA, MASLD may be a concurrent risk factor for iCCA, particularly in populations with established or unidentified underlying chronic liver disease.

The risk of BTC seemed to be greater among participants with reduced diabetes durations and those who did not receive treatment with insulin beyond obesity. These findings encourage us to pay particular attention to non-obese diabetic patients with a brief history of the disease.

However, to thoroughly investigate the underlying pathogenesis of MASLD and CCA, future studies should be based on a histopathological diagnosis of MASH and carried out by skilled hepatobiliary pathologists. Accordingly, early detection and prevention of chronic liver disease, as well as additional intervention studies, will undoubtedly be required to determine whether improvements to MetS, weight loss, and diabetes therapy can reduce the risk and progression of CCA.

Additionally, more large-scale population studies are required to understand the role of reproductive hormones in biliary carcinogenesis and to provide a standardized therapeutic approach to hormonal modification in this patient context.

Ultimately, when approaching BTC patients, it is critical to carefully assess the cardiological risk, as these patients are more likely to have cardiovascular risk factors and are at risk of developing thrombotic complications, cardiovascular death, cardiac metastasis, and nonbacterial thrombotic endocarditis.

## Figures and Tables

**Figure 1 cancers-16-03432-f001:**
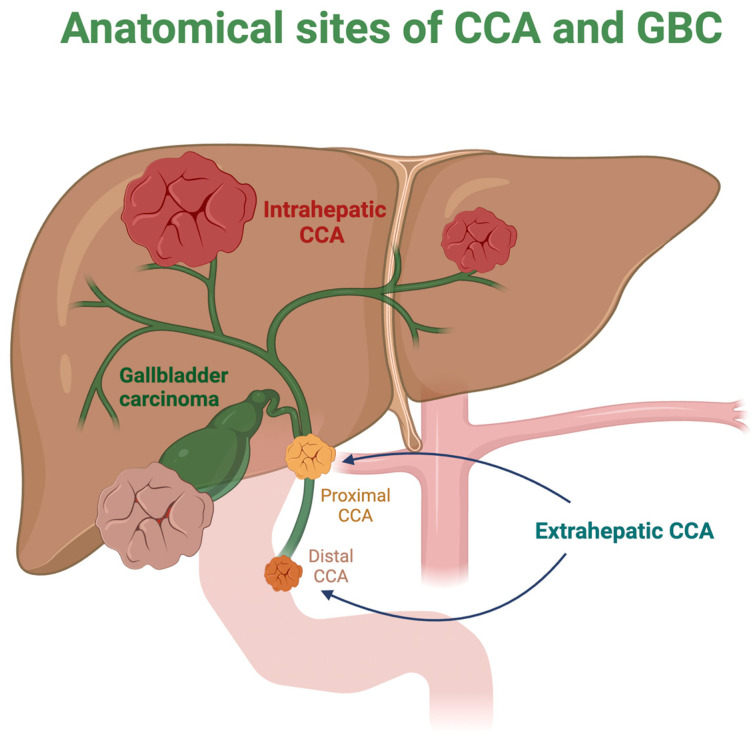
Most prevalent sites of cholangiocarcinoma and gallbladder cancers. CCA: Cholangiocarcinoma; GBC: gallbladder cancer.

**Figure 2 cancers-16-03432-f002:**
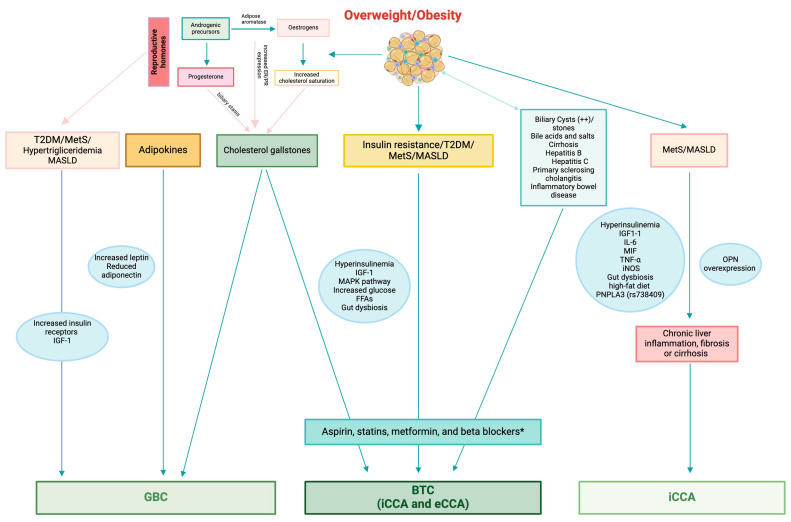
Representation of the link and pathogenic factors that connect MetS, MASLD, T2DM, and other chronic liver diseases to BTC and GBC. Biliary tract cancer: MetS, MASLD, T2DM, and hormonal patterns are associated with BTC, in patients with elevated levels of insulin and IGF-1. Especially patients with shorter diabetes periods and naive to insulin had a greater BTC risk. T2DM patients may be affected by insulin’s anabolic effects, which may promote cholestatic cell proliferation. The MAPK pathway mediates insulin’s mitogenic effect. Choledochal cysts or stones, cirrhosis, hepatitis B, and hepatitis C were the most significant risk factors for both iCCA and eCCA. Estrogens play a significant role in BTC onset, likely due to the conversion function of androgenic precursors into estrogens from the adipose tissue aromatase. Obesity causes an alteration in the production of adipokines, particularly leptin and adiponectin. Leptin receptors are highly expressed in metabolically active tissues, whereas adiponectin levels are low. Consequently, malignant transformation enhances the expression of leptin receptors in normal cholangiocytes. MASLD-associated intestinal dysbiosis may promote carcinogenesis by causing leaky gut, microbiome-associated biological patterns, and bacterial metabolites. (*) Aspirin, statins, metformin, and beta blockers have shown promising pleiotropic properties in preventing BTC even by reducing the risk of thrombotic and cardiovascular complications. Intrahepatic cholangiocarcinoma subtype: MetS/MASLD is associated with iCCA risk. In these patients, the liver produces and releases high levels of proinflammatory cytokines, including insulin, IGF, IL-6, MIF, TNF-α, and iNOS, which can promote carcinogenesis. Mice fed a high-fat diet had significantly more severe cholangitis and iCCA than those fed a standard diet. The risk-increasing allele PNPLA3, rs738409, significantly increased the association between MASLD and iCCA. OPN overexpression may cause biliary carcinogenesis in MetS/MASLD patients with iCCA. These triggers can raise the risk of iCCA due to chronic liver inflammation, fibrosis, or cirrhosis. Extrahepatic cholangiocarcinoma subtype: includes perihilar and distal cancers, and is linked to biliary tract diseases. Diabetes, cholecystolithiasis, chronic pancreatitis, and dyslipidemia were all associated with Vater’s ampulla in eCCA. There is a connection between perihilar CCA and PSC. Gallbladder cancer: MetS/MASLD, T2DM, and hormonal patterns increase even the risk of developing GBC. Elevated serum triglyceride levels may be an independent predictor of GBC risk when combined with gallbladder stone disease, while insulin resistance may be the first factor in GBC that is unrelated to gallbladder stone disease. In vitro studies have shown that GBC has elevated levels of insulin receptors and IGF-1 and that gallbladder tumor cells require continuous insulin stimulation to proliferate. Also, reproductive hormones may play a meaningful role in GBC development. Continuous exposure to high levels of female reproductive hormones during pregnancy can lead to gallstone formation. Many GBC patients exhibit ER and PR receptors. Interestingly, postmenopausal women with higher estradiol levels had a significantly higher risk of iCCA. Early hysterectomy and long-term oral contraceptive use were linked to an increase in iCCA risk. MetS: metabolic syndrome; MASLD: metabolic dysfunction-associated steatotic liver disease; T2DM: type-2 diabetes mellitus; BTC: biliary tract cancer; IGF-1: insulin-like growth factor-1; MAPK: mitogen-activated protein kinase; iCCA: intrahepatic cholangiocarcinoma; eCCA: extrahepatic cholangiocarcinoma; FFAs: fatty-free acids; IL-6: interlekin-6; MIF: macrophage migration inhibitory factor; TNF: tumor necrosis factor; iNOS: inducible nitric oxide synthase; PNPLA3: patatin like phospholipase domain containing 3; OPN: osteopontin; GBC: gallbladder cancer; PSC: primary sclerosing cholangitis; ER: estrogen receptor; PR: progesterone receptor.

**Table 1 cancers-16-03432-t001:** MetS and BTC.

Authors	Year	Study Type	N Patients	Study Period/ Follow up Period (If Applicable)	Risk of BTC	Conclusions
Shebl et al. [16]	2011	Population-based case–control study	627 BTC 959 controls	1997–2001	GBC (OR = 2.75) CCA (OR = 1.92)	MetS and IR are involved in the etiology of BTCs
Welzel et al. [24]	2011	Retrospective population-based study	743 iCCA and 195,953 controls	1994–2005	(aOR = 1.56) *	MetS is a significant risk factor for iCCA in the U.S. population
Xiong et al. [25]	2018	Retrospective hospital-based case–control	136 iCCA and 167 eCCA	2002–2014	iCCA (aOR = 2.68) eCCA (aOR = 1.79) **	MetS is significantly linked to CCA risk
Park et al. [34]	2021	Nationwide population-based cohort study	7506 CCA	Median follow-up of 5.1 years	(aHR = 1.07) ***	MetS can increase the risk of CCA if it persists ≥ 2 years
Zhan et al. [59]	2024	Meta-analyses of 31 prospective studies	N.A.	N.A.	(RR = 1.37)	Lifestyle changes/medical interventions targeting MetS may prevent GBC

MetS: Metabolic Syndrome; BTC: Biliary Tract Cancer; OR: Odd Ratio; HR: Hazard Ratio; RR: Relative Risk; GBC: Gallbladder Cancer; CCA: Cholangiocarcinoma; IR: Insulin Resistance; iCCA = Intrahepatic CCA; U.S.: United States; eCCA: Extrahepatic CCA. * Adjusted for biliary cirrhosis, Cholangitis, cholelithiasis, choledochal cysts, HBV infection, HCV infection, unspecified viral hepatitis, alcoholic liver disease, non-specified cirrhosis, inflammatory bowel disease, Crohn’s disease, ulcerative colitis, smoking. ** Adjusting for age and gender. *** Adjusted for age, gender, smoking status, alcohol consumption, physical activity, body mass index, choledochal cysts, cholangitis, primary biliary cirrhosis, cholelithiasis, cholecystitis, liver flukes, liver cirrhosis, and alcoholic cirrhosis of the liver.

**Table 2 cancers-16-03432-t002:** DM and BTC.

Authors	Year	Study Type	N Patients	Study Period/ Follow Up Period (If Applicable)	Risk of BTC	Conclusions
He et al. [15]	2014	Case–control study	181 patients and 905 controls	2006–2010	Ampullary adenoma (OR = 2.27) Ampullary cancer (OR = 1.98)	DM may contribute to benign ampullary adenomas progressing into cancer
Welzel et al. [18]	2007	Population-based case–control study	535 iCCA 549 eCCA and 102,782 controls	1993–1999	iCCA (OR = 1.8) eCCA (OR = 1.5) *	iCCA and eCCA share some risk factors (including T2DM).
Ren et al. [20]	2011	Meta-analysis of 21 studies (8 case–control and 13 cohort studies)		1974–2010	(summary RRs = 1.43)	There is a link between diabetes and an increased risk of BTC and its subtypes: GBC or eCCA, but not AVC
Schlesinger et al. [21]	2013	Prospective analysis (EPIC-cohort study)	204 BTC (75 GBC and 129 other subtypes)	Mean follow-up of 8.5 years	BTC (RRs = 1.77) GBC (RRs = 2.72)	The research supports the hypothesis that DM increases the risk of BTC (especially GBC and HCC)
Park et al. [22]	2021	Cohort study	13,022 BTC	Median follow-up of 7.2 years	(aHR = 1.31) [newly diagnosed diabetes] (aHR = 1.35) [diabetes duration < 5 years] (aHR = 1.47) [diabetes duration ≥ 5 years] **	Both IFG and DM independently increase the likelihood of BTC (CCA and GBC) A longer duration of DM is associated with a further increase in BTC risk
Petrick et al. [23]	2017	Retrospective study	2092 iCCA 2981 eCCA 323,615 controls	2000–2011	iCCA (OR = 1.54) eCCA (OR = 1.45)	Metabolic conditions are associated with both iCCA and eCCA
Xiong et al. [25]	2018	Retrospective hospital-based case–control	136 iCCA and 167 eCCA	2002–2014	iCCA (aOR = 4.59) eCCA (aOR = 0.97) ***	DM is only related to iCCA
Li et al. [52]	2015	Meta-analysis of twenty studies (fifteen case–control studies and five cohort studies)		1996–2014	CCA (pooled OR 1.74) iCCA (summary RR = 1.93) eCCA (summary RR = 1.66)	DM may increase the risk of CCA
Petrick et al. [54]	2019	Pooled analysis of individual-level data from 13 US-based cohort studies Liver Cancer Pooling Project) and subsequent meta-analysis			(RR = 1.53)	Obesity and DM are associated with increased iCCA risk
Gu et al. [62]	2016	Meta-analysis of 20 studies (8 case–control studies and 12 cohort studies)			(sRR = 1.56)	Compared to non-DM individuals, both men and women with T2DM have a higher risk of GBC

DM: Diabetes Mellitus; BTC: Biliary Tract Cancer; OR: Odd Ratio; HR: Hazard Ratio; RR: Relative Risk; CCA: Cholangiocarcinoma; iCCA= Intrahepatic CCA; eCCA: Extrahepatic CCA; GBC: Gallbladder Cancer; AVC: Ampulla of Vater carcinoma; HCC: Hepatocellular Carcinoma; IFG: Impaired Fasting Glucose; T2DM: Type 2 DM. * Adjusted for age, gender, race, geographic location, and state buy-in status. ** Adjusted for age, gender, body mass index, smoking status, drinking status, physical activity, choledochal cysts, cholangitis, primary biliary cirrhosis, cholelithiasis, cholecystitis, liver flukes, fibrosis and cirrhosis of the liver, and alcoholic cirrhosis of the liver. *** Adjusting for age and gender.

**Table 3 cancers-16-03432-t003:** Obesity and BTC.

Authors	Year	Study Type	N Patients	Study Period/Follow Up Period (If Applicable)	Risk of BTC	Conclusions
Jackson et al. [14]	2019	Pooled data from 27 prospective cohorts with over 2.7 million adults	1343 GBC 1194 eCCA 784 iCCA 623 AVC	37,883,648 person-years of follow-up	GBC (HR = 1.72, 95% CI = 1.41–2.08) iCCA (HR = 2.06) eCCA (HR = 1.33) GBC (HR = 3.32) iCCA (HR = 2.16) No evidence of an association between BMI classification and AVC	There is a correlation between adiposity and BTC
Welzel et al. [18]	2007	Population-based case–control study	535 iCCA 549 eCCA and 102,782 controls	1993–1999	iCCA (OR = 1.7) eCCA (OR = 1.1) *	iCCA and eCCA share some common risk factors (including T2DM). Since MASLD, obesity, and HCV infection were linked to iCCA and are on the rise, they may account for the differing trends in iCCA and eCCA rates.
Grainge et al. [19]	2009	Case–control study	611 BTC (372 CCA, 184 GBC, 55 unspecified BTC)	1987–2002	CCA (OR = 1.52) GBC (OR = 1.51)	DM and obesity increase the risks for BTC
Petrick et al. [23]	2017	Retrospective study	2092 iCCA 2981 eCCA 323,615 controls	2000–2011	iCCA (OR = 1.42) eCCA (OR = 1.17)	Metabolic conditions are associated with both iCCA and eCCA
Xiong et al. [25]	2018	Retrospective hospital-based case–control	136 iCCA and 167 eCCA	2002–2014	iCCA (aOR = 2.13) eCCA (aOR = 1.78) **	Obesity is related to both iCCA and eCCA
Petrick et al. [54]	2019	Pooled analysis of individual-level data from 13 US-based cohort studies Liver Cancer Pooling Project) and subsequent meta-analysis	N.A.	N.A.	(RR = 1.49)	Obesity and DM are associated with increased iCCA risk
Larsoon et al. [57]	2007	Meta-analysis of eight cohort studies and three case–control studies	3288 GBC	1966–2007	(RR = 1.66)	Excess body weight is associated with an increased risk of GBC

BTC: Biliary Tract Cancer; OR: Odd Ratio; HR: Hazard Ratio; RR: Relative Risk; CI: Confidence Interval; GBC: Gallbladder Cancer; CCA: Cholangiocarcinoma; iCCA= Intrahepatic CCA; eCCA: Extrahepatic CCA; AVC: Ampulla of Vater carcinoma; T2DM: Type 2 Diabetes Mellitus; MASLD: Metabolic dysfunction-associated steatotic liver disease; HCV: Hepatitis C virus; HCC: Hepatocellular Carcinoma. * Adjusted for age, gender, race, geographic location, and state buy-in status. ** Adjusted for age and gender.

**Table 4 cancers-16-03432-t004:** MASLD and BTC.

Authors	Year	Study Type	N Patients	Study Period/Follow Up Period (If Applicable)	Risk of BTC	Conclusions
Welzel et al. [18]	2007	Population-based case–control study	535 iCCA 549 eCCA and 102,782 controls	1993–1999	iCCA (OR = 3.0) eCCA (OR = 2.4) *	MASLD, obesity, and HCV infection are associated with increasing iCCA incidence, potentially explaining the divergent iCCA/eCCA rate trends
Petrick et al. [23]	2017	Retrospective study	2092 iCCA 2981 eCCA 323,615 controls	2000–2011	iCCA (OR = 3.52) eCCA (OR = 2.93)	Metabolic conditions are associated with both iCCA and eCCA
Wongjarupong et al. [65]	2017	Meta-analysis (7 case–control studies)	5067 iCCA 4035 eCCA 129.111 controls	Studies published up to April 2017	All CCAs (Pooled OR = 1.97) iCCA (Pooled OR = 2.09) eCCA (Pooled OR = 2.05)	MASLD may increase the risk of CCA progression MASLD has a greater impact on the risk of iCCA compared to eCCA, suggesting a shared pathogenesis between iCCA and HCC
Park et al. [74]	2021	Nationwide cohort study	8,120,674 adults of which 13,043 patients were with newly diagnosed BTC	Median follow-up of 7.2 year	CCA (aHR 1.33) GBC (aHR 1.14)	MASLD is linked to a higher risk of CCA and GBC
Corrao et al. [75]	2021	Meta-analysis	N.A.	N.A.	All CCAs (OR = 1.88) iCCA (OR= 2.19) eCCA (OR= 1.48)	MASLD is only associated with developing iCCA, not eCCA
Liu et al. [79]	2022	Retrospective analysis of large-scale cohort study	352,911 individuals (37.2% with MAFLD), among whom 23,345 developed different types of cancers	N.A.	GBC (HR = 2.20) Liver cancers (HR = 1.81)	Compared with non-MASLD, MASLD is significantly associated with GBC and liver cancers

MASLD: Metabolic Dysfunction-Associated Steatotic Liver Disease; BTC: Biliary Tract Cancer; OR: Odd Ratio; HR: Hazard Ratio; RR: Relative Risk; CI: Confidence Interval; CCA: Cholangiocarcinoma; iCCA: Intrahepatic CCA; eCCA: Extrahepatic CCA; HCV: Hepatitis C virus; HCC: Hepatocellular Carcinoma; GBC: Gallbladder Cancer. * Adjusted for age, gender, race, geographic location, and state buy-in status

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
