# Peer review of "Interplay of Cardiometabolic Syndrome and Biliary Tract Cancer: A Comprehensive Analysis with Gender-Specific Insights"

_cancers, 2024, doi:10.3390/cancers16193432_

Round 1

Reviewer 1 Report

Comments and Suggestions for Authors

Excellent and comprehensive review of the topic. My only suggestion if feasible would be to perhaps briefly comment on any data on prediction/screening of BTC based on metabolomics as effective screening for BTC remains elusive

Author Response

  1. Excellent and comprehensive review of the topic. My only suggestion if feasible would be to perhaps briefly comment on any data on prediction/screening of BTC based on metabolomics as effective screening for BTC remains elusive

R: We are grateful to the author for appreciating our study and for the precious suggestions. We have included a section "Metabolomics and biliary tract cancers screening/prognosis" (page 23) which details the most relevant studies on the screening and prediction of BTC based on metabolomics.

Reviewer 2 Report

Comments and Suggestions for Authors

In this work (Cancers-3199098), Di Stasi et al., reviewed the possible connection between cardiovascular disease and cancers of biliary tract. The study highlights also gender-specific differences. Given the growing incidence of metabolic syndrome and associated disorders this review is interesting for a wide group of readers. However, a few concerns remain.

Concerns:

1.      The authors interchangeably use NAFLD and MASLD, which is confusing. Given that the nomenclature has changed for more than a year ago and is now well accepted in the scientific community it will be better to stick with the present version i.e., MASH and MASLD.

2.      Line 17: The appropriate terminology for writing is ‘reproductive or gonadal hormones’ rather than ‘sexual hormones’.

3.      There are several recent single-cell/nuclei data available on CCAs, there conclusions with the expression of disease relevant pathways as discussed here needs to be integrated.

4.      A figure showing the anatomical places of different CCAs and GBC will help readers with easy visualization and understanding.

5.      The tables shown need to be briefly summarized of their main message. Otherwise its hard to follow.

6.      Regarding gender specificity, is there any indication to whether the cancer incidences rise post-menopause?

7.      The present Fig. 1 quality could be bettered and a clearer legend is needed.

Comments on the Quality of English Language

Moderate changes are necessary to improve readability.

Author Response

1.      The authors interchangeably use NAFLD and MASLD, which is confusing. Given that the nomenclature has changed for more than a year ago and is now well accepted in the scientific community it will be better to stick with the present version i.e., MASH and MASLD.

R: We thank to the Reviewer for his insightful comments. Accordingly, we  have modified within the text the suggested nomenclature as requested.

2.      Line 17: The appropriate terminology for writing is ‘reproductive or gonadal hormones’ rather than ‘sexual hormones’.

R: We modified the terminology.

3.      There are several recent single-cell/nuclei data available on CCAs, there conclusions with the expression of disease relevant pathways as discussed here needs to be integrated.

R: As requested, we presented several studies on the application of single-cell sequencing technology to CCAs and discussed the main results as well as the relevant pathways associated with the disease.

4.      A figure showing the anatomical places of different CCAs and GBC will help readers with easy visualization and understanding.

R: We have added the new Figure 1, as requested.

5.      The tables shown need to be briefly summarized of their main message. Otherwise its hard to follow.

R: We simplified the tables into more readable forms, making the key messages more concise and clear.

6.      Regarding gender specificity, is there any indication to whether the cancer incidences rise post-menopause?

R: We have included in the manuscript the only relevant finding from the literature in this regard (see page 19).

7.      The present Fig. 1 quality could be bettered and a clearer legend is needed.

R: We have improved the quality of the old Fig 1 (see new Fig 2) and included a clearer legend, as requested.

Comments on the Quality of English Language: Moderate changes are necessary to improve readability. R: We have also performed the requested language changes.  Thank you again. 
